# Increased sensitivity in Electron Nuclear Double Resonance spectroscopy with chirped radiofrequency pulses

Julian Stropp[1,*], Nino Wili[2,*], Niels Chr. Nielsen[2], and Daniel Klose[1]

[*]These authors have contributed equally

[1]Institute for Molecular Physical Science, ETH Zurich, Vladimir-Prelog-Weg 2, CH-8093 Zurich, Switzerland.

[2]Interdisciplinary Nanoscience Center and Department of Chemistry, Aarhus University, Gustav Wieds Vej 14, 8000 Aarhus C, Denmark.

**Correspondence:** Niels Chr. Nielsen (ncn@chem.au.dk) & Daniel Klose (daniel.klose@phys.chem.ethz.ch)

**Abstract.** Electron Nuclear Double Resonance (ENDOR) spectroscopy is an EPR technique to detect the nuclear frequency spectra of hyperfine coupled nuclei close to paramagnetic centres, which have interactions that are not resolved in continuous wave EPR spectra and may be fast relaxing on the time scale of NMR. For the common case of non-crystalline solids, such as powders or frozen solutions of transition metal complexes, the anisotropy of the hyperfine and nuclear quadrupole interactions renders ENDOR lines often several MHz broad, thus diminishing intensity. With commonly used ENDOR pulse sequences only a small fraction of the NMR/ENDOR line is excited with a typical RF pulse length of several tens of μs, and this limits the sensitivity in conventional ENDOR experiments. In this work, we show the benefit of chirped RF excitation in frequency domain ENDOR as a simple yet effective way to significantly improve sensitivity. We demonstrate on a frozen solution of Cu(II)-tetraphenylporphyrin that the intensity of broad copper and nitrogen ENDOR lines increases up to 9-fold compared to single frequency RF excitation, thus making the detection of metal ENDOR spectra more feasible. The tunable bandwidth of the chirp RF pulses allows the operator to optimize for sensitivity and choose a tradeoff with resolution, opening up options previously inaccessible in ENDOR spectroscopy. Also, chirp pulses help to reduce RF amplifier overtones, since lower RF powers suffice to achieve intensities matching conventional ENDOR. In 2D TRIPLE experiments the signal increase exceeds 10 times for some lines, thus making chirped 2D TRIPLE experiments feasible even for broad peaks in manageable acquisition times.

## 1   Introduction

Paramagnetic centers are abundant throughout nature and material science and many catalysts involve paramagnetic active species with particular chemical reactivity. (Roessler and Salvadori, 2018; Carter and Murphy, 2015; Hanson and Berliner, 2010; Goldfarb, 2022) To garner molecular information on paramagnetic sites of interest, including on the geometric and electronic structures, Electron Paramagnetic Resonance (EPR) spectroscopy is a sensitive and widely used technique to identify the paramagnetic species, the spin states and, in case of metal ions, the oxidation state and ligand field. (Roessler and Salvadori, 2018; Bonke et al., 2021) Particularly the hyperfine (and quadrupole for $I > \frac{1}{2}$) couplings to nuclei with spin $I$ provide interesting structural and chemical information often sought after to reveal molecular details. (Schweiger and Jeschke, 2001;

Pilbrow, 1990; Formanuik et al., 2016; Allouche et al., 2018; Ashuiev et al., 2021) To resolve the electron-nuclear interactions, also below the inhomogeneous EPR line width, pulse EPR hyperfine spectroscopy is an important toolkit represented by a variety of methods that have previously been reviewed. (Roessler and Salvadori, 2018; Harmer, 2016; Van Doorslaer, 2017; Goldfarb, 2017; Wili, 2023)

Out of these hyperfine techniques, the class known as pulse Electron-Nuclear Double Resonance (ENDOR) experiments makes use of double excitation of electron spins and nuclear spins by microwave (MW) and radiofrequency (RF) pulses, respectively. (Feher, 1956; Harmer, 2016) ENDOR has the potential to provide high-resolution nuclear frequency spectra and features Pake patterns as line shapes that can directly reflect the anisotropy of the electron-nuclear interaction tensors. Of the two most widely applied pulse ENDOR experiments, Davies ENDOR (Davies, 1974) is most suited for this purpose, since it only features a central blindspot at the nuclear Larmor frequency. This diminishes the intensity of very small hyperfine couplings, but otherwise does not significantly distort the line shapes of peaks from nuclei with stronger hyperfine couplings. Davies ENDOR relies on an initial MW pulse, which selectively inverts a single EPR transition, subsequent electron-nuclear polarization transfer onto an NMR transition by an RF pulse, followed by echo detection of the remaining electron spin polarization on the EPR transition. Thus ENDOR shows the nuclear frequency spectrum via changes in the electron spin echo intensity. (Harmer, 2016) Mims ENDOR (Mims, 1965) is more sensitive to detect smaller couplings, yet suffers from periodic blindspots in the ENDOR spectrum due to the polarization grating generated by the initial $(\pi/2) - \tau - (\pi/2)$ preparation block, accordingly summation over spectra with a suitable range of $\tau$ delays is used to eliminate blind spots.

ENDOR experiments have been performed in two dimensions by extending the nuclear frequency spectra with a second indirect dimension either based again on the ENDOR effect in 2D TRIPLE ENDOR (Mehring et al., 1987; Epel and Goldfarb, 2000), or by using a frequency-selective hole-burning pulse in THYCOS (Potapov et al., 2008), or by an additional evolution period to correlate nuclear frequencies in the different electron spin manifolds, which is analogous to HYSCORE, in the HYEND experiment (Jeschke and Schweiger, 1995a).

The advantages of the ENDOR experiments, however, come at the price that double MW and RF excitation schemes are technically more demanding and the typically long RF pulses require sufficiently long electron spin relaxation times. (Harmer, 2016) Also important to consider is nuclear spin relaxation. When this becomes relevant on the time scale of the ENDOR sequences, it can at times lead to spectral distortions. When nuclear relaxation is instead slow with respect to the experimental repetition time, nuclear saturation effects may be observed and are typically alleviated by stochastic RF excitation of the frequency-domain ENDOR spectra. (Rizzato and Bennati, 2014) A challenge often encountered on transition metal complexes is that sensitivity becomes limiting instead of spectral resolution because the ENDOR lines are typically broader than the limited RF excitation bandwidth achievable with single frequency pulses (Harmer, 2016), unless microresonators with specialized coils are employed. (Dayan et al., 2022) Therefore, under common conditions the number of spins excited and detected in orientation-selective Davies ENDOR experiments is small and thus limits sensitivity.

Here, we address the challenge of improving sensitivity in frequency domain ENDOR experiments by introducing frequency-swept RF pulses. These pulses have been used for a long time in NMR for adiabatic passages of nuclear transitions and their pulse shapes have been optimized over the last decades. (Abragam, 1961; Baum et al., 1985; Garwood and DelaBarre, 2001;

Kupce and Freeman, 1996) In ENDOR spectroscopy such pulses have been previously used in time domain ENDOR se-
quences on sharp resonance lines in crystals. (Jeschke and Schweiger, 1995b) These time domain experiments have so far not
found widespread application, probably because sufficient excitation bandwidth is technically challenging. Here, using chirp
RF pulses instead in frequency-domain ENDOR to broaden the RF excitation bandwidth, we demonstrate on a Cu(II) model
system that, depending on the ENDOR line width, significant sensitivity is gained in chirp Davies and chirp Mims ENDOR
experiments. This not only enhances the spectral intensity of commonly observable nuclei, [1]H and [14]N, up to five-fold com-
pared to single frequency ENDOR, but also renders hard-to-observe nuclei such as the central metal ions spectroscopically
accessible, which is demonstrated here for [63,65]Cu. Overly broad RF excitation bandwidths lead to broadening of the ENDOR
spectrum and thus decrease spectral resolution. Since the gain in sensitivity in this trade-off is general for the ENDOR dimen-
sion, also in multi-dimensional hyperfine spectroscopy, we demonstrate the advantage for Mims ENDOR and, particularly, for
2D TRIPLE. Thus, we show how chirp-RF excitation can significantly enhance the ENDOR dimension used in different 2D
pulse hyperfine techniques, especially for broad resonance lines.

## 2   Materials and methods

### 2.1   Sample Preparation

The 2 mM CuTPP sample was prepared by dissolving Copper(II) tetraphenylporphyrin (abcr GmbH, Karlsruhe, Germany) in
a 1:1 mixture of fully deuterated dichloromethane, $CD_2Cl_2$ (Cambridge Isotope Laboratories Inc., Andover, USA), and $d_8$-
toluene (Sigma-Aldrich Chemie GmbH, Steinfelden, Germany). 40 µl of the solution were transferred into a quartz tube with
3 mm outer diameter and frozen in liquid nitrogen.

### 2.2   EPR measurements

Pulse EPR data were acquired in X band on a homebuilt AWG-based pulse EPR spectrometer equipped with an MD-4 ENDOR
resonator (Bruker BioSpin, Ettlingen, Germany) and a TWT amplifier with a nominal output power of 1 kW. The RF pulses
were generated with a separate AWG (HDAWG, Zurich Instruments, Zurich, Switzerland) and amplified with a 500 W RF
amplifier (Amplifier Research Inc., Souderton PA, USA). Before the probehead a diode circuit was mounted to reduce the RF
noise level around RF pulses; after the probehead the RF circuit was terminated by a load. The spectrometer is equipped with a
cryogen-free variable temperature EPR cryostat (Cryogenic Ltd., London, UK) to maintain a stable sample temperature of 15
K during the experiments.
All ENDOR spectra were recorded at the maximum of the echo-detected EPR spectrum at 340.5 mT, a MW frequency of 9.78
GHz and a shot repetition time of 10 ms. Davies ENDOR spectra were acquired with a rectangular 200 ns selective microwave
pulse and observer $\pi/2$ and $\pi$ pulses of 10/20 ns with $\tau = 420$ ns. For Mims ENDOR, 10 ns $\pi/2$ pulses were used with a
$\tau$ value of 420 ns. The ENDOR spectra were recorded with stochastic acquisition, with RF pulses of either 100 or 500 W
amplifier output power. RF pulses were separated by a 1 µs predelay and a 5 µs postdelay from the MW pulses to reduce RF

ringing. The edges of the chirp RF pulses were shaped with quarter sine waves in the first and last 200 ns. For both Mims and Davies ENDOR a 4-step phase cycle was used with 25 scans and the full width of the echo was integrated. The phase cycle for Davies ENDOR was $\pi(0,0,0,0) - p_{\mathrm{RF}}(0,0,0,0) - \pi/2(0,0,\pi,\pi) - \pi(0,\pi,0,\pi) - \mathrm{Detection}(1,1,-1,-1)$ and for Mims ENDOR $\pi/2(0,\pi,0,\pi) - \pi/2(0,0,\pi,\pi) - p_{\mathrm{RF}}(0,0,0,0) - \pi/2(0,0,0,0) - \mathrm{Detection}(1,-1,-1,1)$. Davies ENDOR spectra were offset corrected with the mean echo intensity between 92 to 95 MHz. For comparison of different ENDOR intensities, peak intensity values at different spectral positions were extracted after this offset correction. The 2D TRIPLE experiment was performed with two 40 µs chirp RF pulses (100 W amplifier output) and a 1 µs delay in between with the otherwise unchanged Davies ENDOR sequence described before (125 scans/2.7 days acquisition time). Three selected chirp TRIPLE traces were recorded with for a longer time (750 scans/30 min acquisition time per trace) and compared to single frequency TRIPLE traces with 8 µs single frequency RF pulses of 100 W. In all TRIPLE traces, the second RF pulse was stepped linearly through the ENDOR spectrum. The phase cycling in TRIPLE experiments was the same as for Davies ENDOR. Nutation experiments were performed with the Davies ENDOR sequence described above by incrementing the RF pulse length at a fixed RF frequency, while keeping the MW pulse sequence and timings constant.

## 2.3 Chirp ENDOR simulations

ENDOR spectra with different RF chirp pulses were simulated for an electron ($S = 1/2$) and a proton ($I = 1/2$) with a Gaussian distribution of purely isotropic hyperfine couplings centered at $A_{\mathrm{mean}} = 4$ MHz with a standard deviation $\sigma = 0.5$ MHz. The relative probability $p(A_i)$ of a specific hyperfine coupling $A_i$ is given by

$$p(A_i) = \frac{1}{\sqrt{2\pi\sigma^2}} \cdot e^{-\frac{(A_i - A_{\mathrm{mean}})^2}{2\sigma^2}} \ . \tag{1}$$

The evolution of the spin density operator during the pulse sequence was simulated for each isotropic hyperfine coupling $A_i$ of the Gaussian distribution as an independent electron-nuclear 2-spin system. A perfect selective inversion pulse on one electron spin transition is assumed, which creates longitudinal 2-spin order, i.e. a $2\hat{I}_z\hat{S}_z$ state. The spin density evolution starting from the $2\hat{I}_z\hat{S}_z$ state is simulated during the chirped RF pulse with the Hamiltonian in linear frequency units

$$\hat{H}_i = \nu_H \hat{I}_z + A_i \hat{I}_z \hat{S}_z + \nu_2 \hat{I}_x \tag{2}$$

using the Liouville-von-Neumann equation with time steps equal to the RF AWG sampling period. $\nu_H$ is the proton Larmor frequency at X band (14.1 MHz) and $\nu_2$ is the amplitude function of the RF pulse with quarter sine edges (see SI section 1). Since the echo signal is proportional to $\langle \hat{I}_\alpha \hat{S}_z \rangle$ after the RF pulse, we use the population difference between the $\alpha_S\alpha_I$ and $\beta_S\alpha_I$ states as signal intensity. For a specific 2-spin system with the isotropic hyperfine coupling $A_i$ and a certain chirp center frequency $\nu_{\mathrm{RF}}$ the signal intensity is

$$I(A_i, \nu_{\mathrm{RF}}) \propto \langle \hat{I}_\alpha \hat{S}_z \rangle \ . \tag{3}$$

The total ENDOR signal at a certain chirp center frequency, $\nu_{\mathrm{RF}}$, is then given by the sum of $I(A_i, \nu_{\mathrm{RF}})$ for the different isotropic hyperfine couplings weighted by their relative probability $p(A_i)$ according to their Gaussian distribution, thus

$$I_{\mathrm{ENDOR}}(\nu_{\mathrm{RF}}) = \sum_{i=1}^{n} I(A_i, \nu_{\mathrm{RF}}) \cdot p(A_i) . \tag{4}$$

The ENDOR spectrum is constructed by calculating $I_{\mathrm{ENDOR}}(\nu_{\mathrm{RF}})$ for the whole RF frequency range. In this simulation, implemented in Matlab (The MathWorks Inc., Natick, MA), relaxation effects are neglected and MW pulses are assumed to be ideal.

The chirp ENDOR spectrum was further reproduced by convolution of an unbroadened, experimental single-frequency (sf) spectrum and the RF chirp pulse excitation profile. The profile for each chirp bandwidth was calculated in EasySpin from the RF waveform shape (see SI section 1 for parameters of the waveform) for frequency offsets from the chirp center frequency $\nu - \nu_{\mathrm{RF}}$. (Stoll and Schweiger, 2006; Pribitzer et al., 2016) A frequency-independent peak amplitude of 50 kHz was used, which corresponds to a sf $\pi$ pulse length of 10 µs. This excitation profile $E(\nu - \nu_{\mathrm{RF}})$ was used in a discrete convolution with the 8 µs sf ENDOR spectrum $I_{\mathrm{sf}}$ to obtain the broadened ENDOR spectrum $I_{\mathrm{sim,chirp}}$:

$$I_{\mathrm{sim,chirp}}(j) = (I_{\mathrm{sf}} * E)(j) = \sum_{m=-M}^{M} I_{\mathrm{sf}}(j - m) E(m) . \tag{5}$$

The discrete convolution requires that the spectrum and the excitation profile have the same frequency resolution, which was 0.1 MHz. Indices $j$, $m$ and $j - m$ indicate the position in the spectrum or excitation profile and $\pm M$ correspond to $\pm 5$ MHz offset in the excitation profile. After the convolution the simulated spectra for different chirp bandwidths were normalized and thus intensity-matched with the corresponding normalized chirp ENDOR spectrum.

## 3 Results and discussion

### 3.1 Single frequency and chirp RF pulses in ENDOR

The benefit of chirped RF pulses in Davies and Mims ENDOR (Fig. 1a and 1b) is shown using the well-studied model sample Cu(II) tetraphenylporphyrin (CuTPP). (Greiner et al., 1992; Brown and Hoffman, 1980; Shao et al., 2005; Van Willigen and Chandrashekar, 1986) Whereas in the ENDOR spectra of CuTPP found in literature (Shao et al., 2005; Greiner et al., 1992; Van Willigen and Chandrashekar, 1986) only protons with small hyperfine couplings and strongly coupled nitrogens were detected, the Davies ENDOR spectrum of CuTPP in Fig. 1c shows additional peaks at RF frequencies above 30 MHz, which can be assigned to the copper hyperfine coupling. Together with the proton and nitrogen peaks, these ENDOR features with different resolutions, intensities and coupling regimes make the sample ideal to test chirp RF pulses in ENDOR experiments.

Using RF pulses with powers of up to 500 W, limited by the RF amplifier output, allows here for 3.5 µs RF-$\pi$ pulses on the strongly coupled $^{14}$N and 8 µs $\pi$ pulses on $^{1}$H, although large amplifier overtones become visible under these conditions (see Fig. S2). With 100 W amplifier output power the RF $\pi$ pulse length increases to 8 µs for $^{14}$N and 15 µs for $^{1}$H, with the advantage that the amplifier does not generate any visible overtones. The difference in $\pi$ pulse lengths at different RF

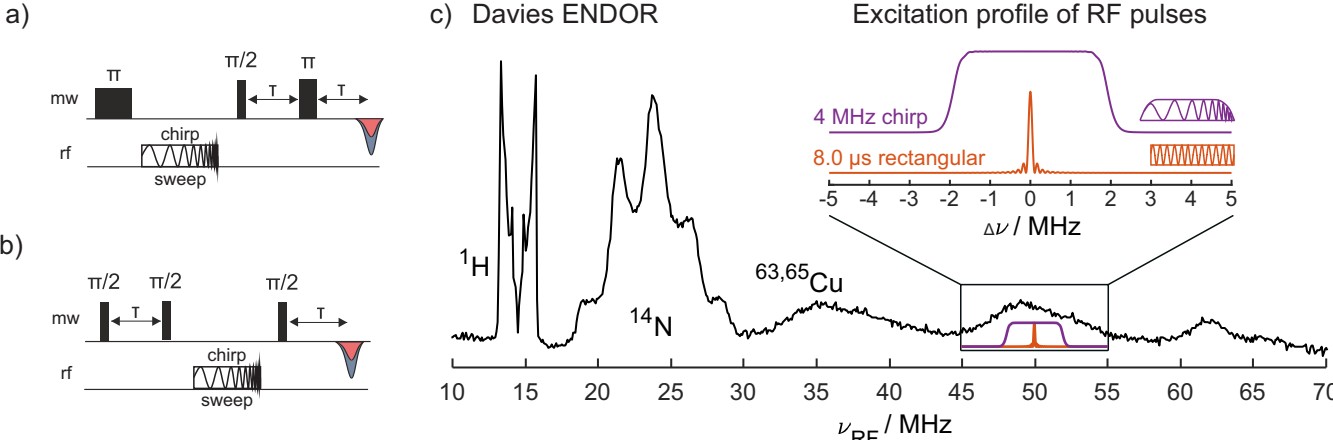

**Figure 1.** a) Davies ENDOR pulse sequence with a chirp RF pulse; b) Mims ENDOR pulse sequence with a chirp RF pulse; c) Davies ENDOR spectrum of CuTPP with a 8 μs single frequency (sf) RF pulse ($\pi$ pulse on strongly coupled $^{14}$N at 100 W RF power) and a selective MW $\pi$ pulse of 128 ns. The inset shows the excitation bandwidth of two types of RF pulses: an 8 μs rectangular single frequency pulse (orange) as used for this Davies ENDOR spectrum and a 4 MHz wide chirp pulse with quarter sine weighted edges (purple; edges span 2 μs of 80 μs total pulse length) calculated with EasySpin. (Stoll and Schweiger, 2006)

frequencies is assigned to the incomplete compensation of the hyperfine enhancement by the $1/\nu_{RF}$ field strength dependence of $B_2$ and different transition moments for nuclei with $I > 1/2$. (Harmer, 2016) The resulting RF excitation bandwidths (FWHM) of the rectangular 3.5 μs and 8 μs pulses are 0.22 MHz and 0.10 MHz, respectively. (Schweiger and Jeschke, 2001) These bandwidths of the rectangular single frequency RF pulses are much smaller than the width of some peaks in the CuTPP ENDOR spectrum (e.g. $^{14}$N and $^{63,65}$Cu, Fig. 1c). Hence, the exchange of the single frequency RF pulse with a linearly frequency-swept RF pulse (called chirp pulse) increases the excitation bandwidth. If enough RF power is available for complete inversion, the ENDOR signal intensity increases considerably. Furthermore, with an arbitrary waveform generator as an RF source, also the rectangular RF pulses can be modified to have quarter sine edges to remove wiggles in the excitation profile of the RF pulse (Fig. 1c).

### 3.2 Chirp ENDOR performance and experimental optimization

The signal increase with chirped RF pulses can be expected to depend on RF pulse power, pulse length and bandwidth as well as the ENDOR line of interest due to its width and relaxation properties. To investigate these dependencies, ENDOR spectra with several pulse lengths and chirp bandwidths were recorded and compared to the corresponding single frequency ENDOR spectrum (Fig. 2 and SI Figs. S3, S4, S5). Details on how to set up and optimize chirp ENDOR experiments can be found in the SI section 2. An increase in the chirp bandwidth increases the intensity of the ENDOR lines without deteriorating the resolution as long as the bandwidth is smaller than the width of the spectral features (Fig. 2a), also clearly visible in the normalized ENDOR spectra (SI Fig. S3b). If both are of the same width, the ENDOR signal still increases, yet at the tradeoff

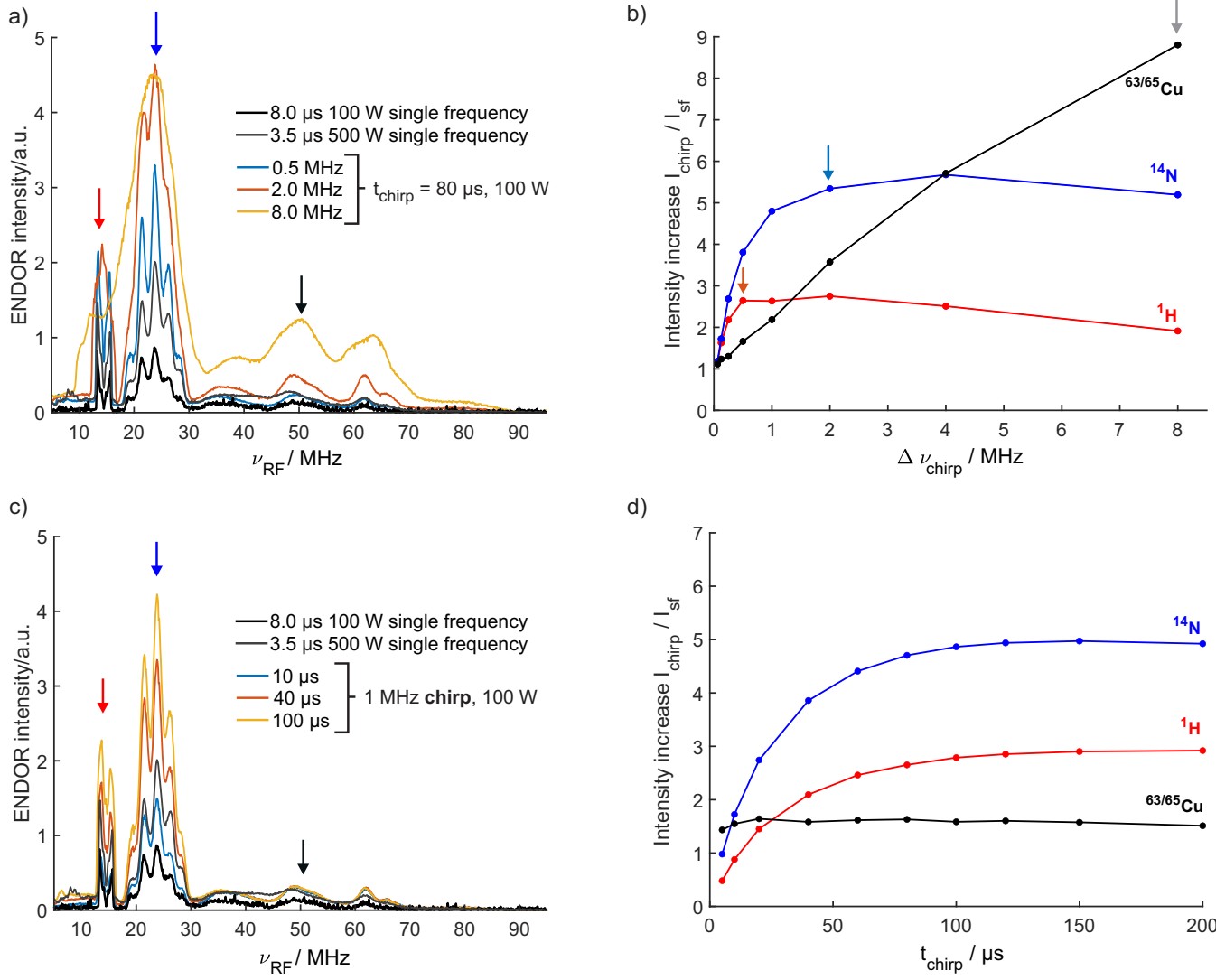

**Figure 2.** Davies ENDOR spectra of CuTPP with chirped RF pulses of (a) different bandwidths (80 μs RF pulse length) and (c) different pulse lengths (1 MHz chirp bandwidth) compared to single frequency (sf) ENDOR spectra. Arrows indicate ENDOR intensities of largest [1]H, [14]N and [63,65]Cu peaks in red, blue, black, respectively, that are quantified in b) and d). Relative ENDOR intensity increase of the largest [1]H, [14]N and [63,65]Cu peaks in chirp ENDOR spectra compared to single frequency ENDOR spectrum (100 W) for different RF bandwidths (b) and different RF pulse lengths (d). Arrows in b) mark RF chirp bandwidths showing a visible onset of broadening for [1]H (0.5 MHz), [14]N (2.0 MHz), and [63,65]Cu (8 MHz) in pale blue, pale red and gray, respectively.

of a loss in resolution (see arrows in Fig. 2b). When the bandwidth is larger than the ENDOR line width, the intensity starts to decrease since the spectral power density gets smaller and the ENDOR transitions are only partially excited. Accordingly,

we find the sharp proton peaks to increase up to an RF chirp bandwidth of 0.5 MHz, whereas for the broad copper peaks signal enhancements of a factor of 9 compared to the single frequency (sf) ENDOR spectrum are achieved at a much larger

bandwidth of 8 MHz (Fig. 2b). Figures 2c and d show that the proton and nitrogen ENDOR sensitivity increases asymptotically to a maximum with longer chirp pulses. This can be explained by the slower frequency sweep rate through the fixed bandwidth of the chirp pulse for longer pulses, which enables a more adiabatic passage of the nuclear spins and a better inversion efficiency. (Baum et al., 1985; Doll et al., 2013) At a chirp bandwidth of 1 MHz a pulse length of about 100 μs is sufficient to achieve maximum intensity as visible in Fig. 2c and d. The length of a 1 MHz chirp RF pulse does not have a significant influence

on the line intensity as long as a sufficient pulse length is used for inversion. Supposedly, chirp pulses with larger bandwidths require a longer pulse length to reach the maximally achievable signal increase. A limiting factor here are the RF coils in the commercial pulsed ENDOR resonator, since they can only handle maximum pulse powers for a limited time, on the order of a few hundred μs. For all tested chirp pulse bandwidths and lengths the signal of all ENDOR lines is at least as high as for the single frequency ENDOR spectrum with the same RF power (100 W). The comparison with the 500 W single frequency

ENDOR spectrum shows, that chirp ENDOR with 100 W RF power is superior above a certain chirp width in terms of signal intensity and is found free from the [14]N amplifier overtones, which are visible in the 500 W spectrum at around 8 MHz (SI Fig. S2). These artificial lines from higher harmonics of the RF amplifier output occur at high RF powers, when the shortest RF pulses are used for maximum ENDOR sensitivity. They are best avoided, since they may corrupt the ENDOR spectrum. To this end, chirp ENDOR experiments deliver the possibility to reduce RF power without sacrificing sensitivity. It is also possible to

use chirp RF pulses with the full RF power of 500 W, although the amplifier overtones become much more pronounced. The trends at 500 W regarding chirp bandwidth and pulse length are similar to the results obtained at 100 W (SI Fig. S4).

Chirp RF pulses were also tested in the Mims ENDOR experiment, which is commonly used to determine small hyperfine and quadrupole couplings and, therefore, usually a higher resolution is required than in Davies ENDOR. (Harmer, 2016) Figure S6 shows that for CuTPP a signal increase of 3 to 4 times can be achieved for the nitrogen and copper lines, whereas for protons

in CuTPP the maximal ENDOR efficiency is reached already at small chirp bandwidths, resulting in a complete echo decay. As expected, smaller chirp bandwidths are required in order to maintain narrower line shapes, and due to this the sensitivity increase is smaller compared to what we observed in Davies ENDOR.

With some prior knowledge about ENDOR line widths or a good initial guess for the chirp bandwidth, chirp RF pulses can help to acquire ENDOR spectra faster without loss of resolution (as discussed below). For very broad features (e.g. couplings

to metal centers) chirp ENDOR can render the measurement feasible within a reasonable time frame.

### 3.3   Chirp ENDOR simulations and spectral convolution

The experimental results are supported by spin dynamics simulations of the chirp ENDOR experiment on an electron-proton system with a Gaussian distribution of isotropic hyperfine couplings ($\sigma = 0.5$ MHz). The simulated proton ENDOR spectra with a mean isotropic hyperfine coupling of 4 MHz are shown in Fig. 3a for different chirp bandwidths. The full width at

half maximum of the intrinsic ENDOR line is 1.2 MHz. As seen in the experimental spectra, simulated ENDOR spectra with chirp bandwidths smaller than this value are only intensified, but not broadened. For larger chirp bandwidths the spectra are

broadened and may decrease in absolute intensity, since the spectral power density becomes too low to fully invert an ENDOR transition for an RF field strength of 100 kHz. Simulations with a 10 times higher RF field strength (1 MHz) show that the decrease does not occur for 40 μs chirp pulses of up to 8 MHz (Fig. S7). If higher RF powers are used in experiments, shorter chirp pulse lengths become possible. This might be interesting for samples with fast relaxing paramagnetic sites. For very large chirp bandwidths, which affect both ENDOR transitions (4 and 8 MHz chirps), the reduction in excitation efficiency due to a lower spectral power density can be partially compensated by excitation of both NMR transitions within the same chirp pulse. For the 8 MHz chirp this leads to a spectrum with two steps and the highest intensity is achieved at an RF frequency in between the two intrinsic ENDOR lines, as seen in Fig. 3. In contrast to the 4 and 8 MHz chirp ENDOR spectra here, the excitation of both coupled NMR transitions has been exploited before with two separate single frequency RF pulses in special TRIPLE experiments in a favorable, quantitative fashion. (Dinse et al., 1974; Epel et al., 2003) In chirp ENDOR the double excitation with a single chirp RF pulse is unwanted since it complicates the spectrum and should be avoided.

Spin dynamics simulations provide valuable insights into the chirp ENDOR experiment, but they become infeasible for spectral analysis of larger spin systems, as CuTPP, due to the dramatic increase in computational cost for time-domain simulations with increasing numbers of spins, (Kuprov et al., 2007) and a simpler simulation approach becomes necessary. The spectra in Fig. 3b are calculated by convolution of the 8 μs single frequency ENDOR spectrum with the respective chirp pulse excitation profile. For both proton and nitrogen peaks the convoluted spectra match well with the experimental spectra, thus providing a viable data analysis approach. Minor deviations are to be expected: First, a constant pulse amplitude is used for the excitation profile and frequency dependencies from the RF amplifier and coil are neglected. Second, the chirp pulse might affect multiple ENDOR transitions in the same electron spin manifold, which is especially relevant here for $I > 1/2$, as for $^{14}$N. Transitions that have an energy level in common will interfere with each other during the passage of the nuclear spin transitions by the chirp RF pulse. (Doll and Jeschke, 2017; Jeschke et al., 2015) In such cases, the convolution approach is not expected to result in an accurate line shape. Third, the convolution does not take into account that the single frequency spectrum itself might already be broadened due to the sinc excitation profile of the rectangular RF pulse. Despite the limitations mentioned, Fig. 3b demonstrates that chirp ENDOR spectra can be well reproduced by the convolution approach. This shows that chirp ENDOR spectra can be analyzed using frequency-domain simulations to obtain the unbroadened spectrum and subsequent convolution with the chirp pulse excitation bandwidth to compute the experimental broadened spectrum. Fitting spin Hamiltonian parameters using chirp ENDOR spectra is thereby feasible in a manner analogous to using single frequency ENDOR data. Yet, for large chirp bandwidths the resolution lost at the benefit of a signal increase cannot be artificially brought back. Hence, the experiment can be optimized by tuning the tradeoff either for more resolution with smaller chirp bandwidths or for more signal intensity with larger chirp bandwidths.

## 3.4 2D TRIPLE experiments

The maximum signal increase observed in 1D Davies ENDOR with chirped RF pulses is 2.5 times for protons, 5 times for nitrogen, and up to 9 times for copper. An even higher signal increase can be obtained in TRIPLE experiments, which use two RF pulses (pulse sequence shown in SI Fig. S8a). For a 1D TRIPLE spectrum the frequency of the first RF pulse is kept

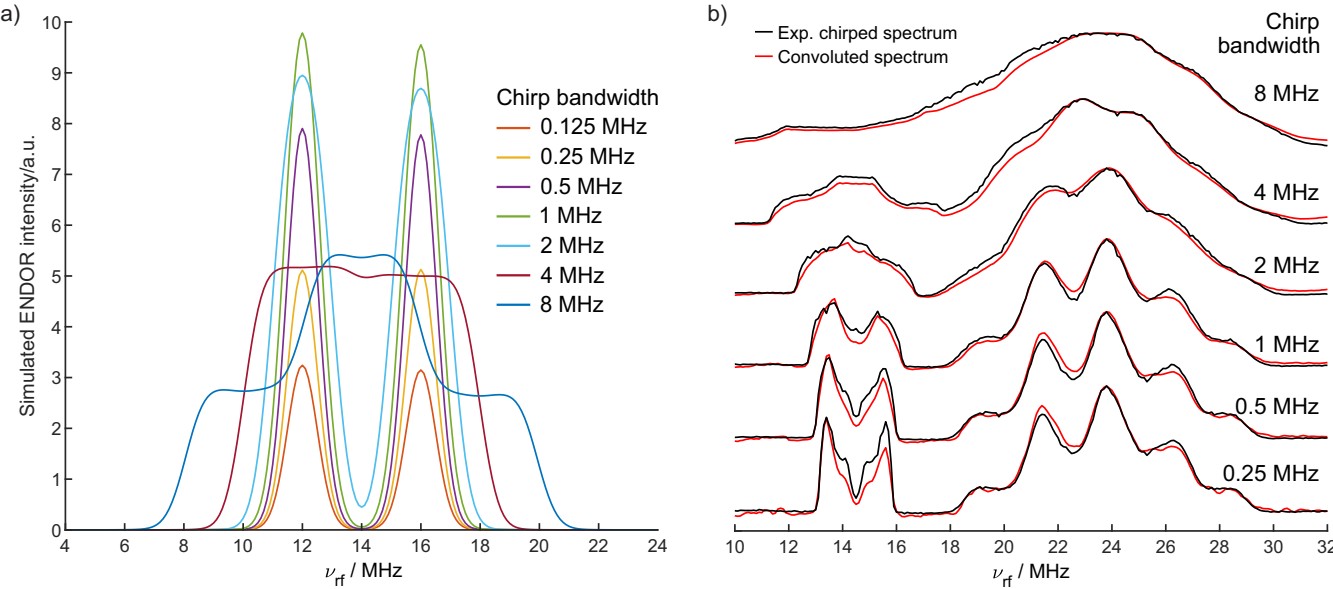

**Figure 3.** a) Simulated chirp ENDOR spectra for different RF chirp bandwidths of an electron proton spin system with a Gaussian distribution of hyperfine couplings ($\sigma = 0.5$ MHz). The RF chirp pulse has length of 40 µs with 200 ns quarter sine wave weighted edges and an RF amplitude of $\nu_{2,\mathrm{max}} = 100$ kHz. b) Comparison of experimental chirp Davies ENDOR spectra (black) with spectra calculated by convolution of the 8 µs single frequency ENDOR spectrum with the excitation profiles of chirp pulses with different bandwidths (red).

constant while the frequency of the second RF pulse is stepped. In the TRIPLE difference spectrum, obtained by subtraction of the ENDOR spectrum from the TRIPLE spectrum, the NMR transitions in the same electron spin manifold are visible. Hence, the spectra aid in assigning different transitions in complex systems and allow for the determination of the relative sign of the hyperfine coupling. (Biehl et al., 1975; Mehring et al., 1987) The 2D difference TRIPLE spectrum is obtained by additionally incrementing the frequency of the first RF pulse and simplifies congested spectra by spreading them along two dimensions at the cost of much longer measurement times. (Epel and Goldfarb, 2000)

The 2D chirp TRIPLE difference spectrum of CuTPP in Fig. 4a shows correlations between proton, nitrogen and also with very weak copper peaks, which would be infeasible to detect using single frequency pulses in 2D TRIPLE experiments even with long acquisition times. In the 2D spectrum the NMR transitions in the same electron spin manifold can be clearly identified for all three coupled types of nuclei. This separation reduces the number of peaks along one dimension by a factor of 2 and the eight overlapping nitrogen NMR transitions are resolved and can be assigned (see zoom inset in Fig. 4a). For this purpose a single chirp RF pulse should not excite two RF transitions from different electron spin manifolds, which is why chirp bandwidth of 0.5 MHz was chosen as a compromise between gain in signal intensity and necessary resolution. Selected TRIPLE traces recorded with single frequency and chirped RF excitation are compared in Fig. 4b and Tab. 1 to estimate the intensity increase and saving of measurement time (chirp and single frequency TRIPLE & ENDOR spectra are shown in Fig.

S8). The signal intensity increase in the ENDOR experiment with a 40 µs chirped RF pulse of 0.5 MHz bandwidth is 1.6x for copper, 2.2x for protons and 3.3x for nitrogen (Fig. S5). In the chirp TRIPLE difference traces the intensity of the triplet line excited with $\nu_{RF,1}$ is increased by a factor of 2.0 for copper, 3.5 for protons and 3.7 for nitrogen compared to TRIPLE experiments with single frequency pulses (see Tab. 1). In principle, the intensity increase observed in TRIPLE should be that

of ENDOR experiments to the power of 2 because two chirped RF pulses are required. Since in TRIPLE difference traces the intensity of neighboring peaks is changed and the peaks are not clearly separated, the intensity of the peaks and not the integral was compared, leading to a slightly lower increase compared to the simple expectation. More importantly, Tab. 1 shows that the signal of NMR transitions connected to the initially excited transition increases up to 12.9 times. The average intensity increase of peaks analyzed in Tab. 1 is 5.7 times, which is equivalent to a measurement time reduction of 32.5 times from

estimated 88.8 days down to 2.7 days for the full 2D TRIPLE experiment (see Fig. 4a). The measurement time was further reduced by using non-uniform frequency steps in both sweep dimensions, i.e. proton and nitrogen peaks were recorded in steps of 0.1 MHz for the chirp center frequency, whereas copper peaks and baseline were recorded in 1 MHz steps. This overall measurement time reduction might turn 2D TRIPLE into a more commonly used experiment, which before has rarely been employed for disordered solids because of the long acquisition times. (Goldfarb et al., 2004; Niklas et al., 2009) An additional

gain in sensitivity is possible in the future by adjusting the bandwidth of the chirp pulses to the ENDOR peak width (i.e. using 8 MHz for copper peaks versus 2 MHz for nitrogen peaks). As an optimal reference for this case, the ENDOR spectrum will then also be measured with the same non-uniform bandwidth excitation scheme as the TRIPLE to obtain a well-defined TRIPLE difference spectrum. While technically feasible, the quantitative information of peak intensities among different coupled nuclei in the ENDOR spectra might become compromised, which remains to be tested in further studies.

**Table 1.** Intensity increase of three chirp TRIPLE difference traces ($\nu_{RF,1}$) at five selected frequency positions ($\nu_{RF,2}$) compared to corresponding single frequency TRIPLE resonance traces with the same number of scans; for traces see Fig. 4b.

| $\nu_{\mathrm{RF,1}}$ \ $\nu_{\mathrm{RF,2}}$ | 13.4 MHz | 15.6 MHz | 21.6 MHz | 23.8 MHz | 49.0 MHz |
|---|---|---|---|---|---|
| 13.4 MHz | 3.5 | 4.0 | 9.8 | - | 4.1 |
| 23.8 MHz | 10.2 | 8.0 | 12.9 | 3.7 | - |
| 49.0 MHz | 3.0 | - | 4.1 | 3.3 | 2.0 |

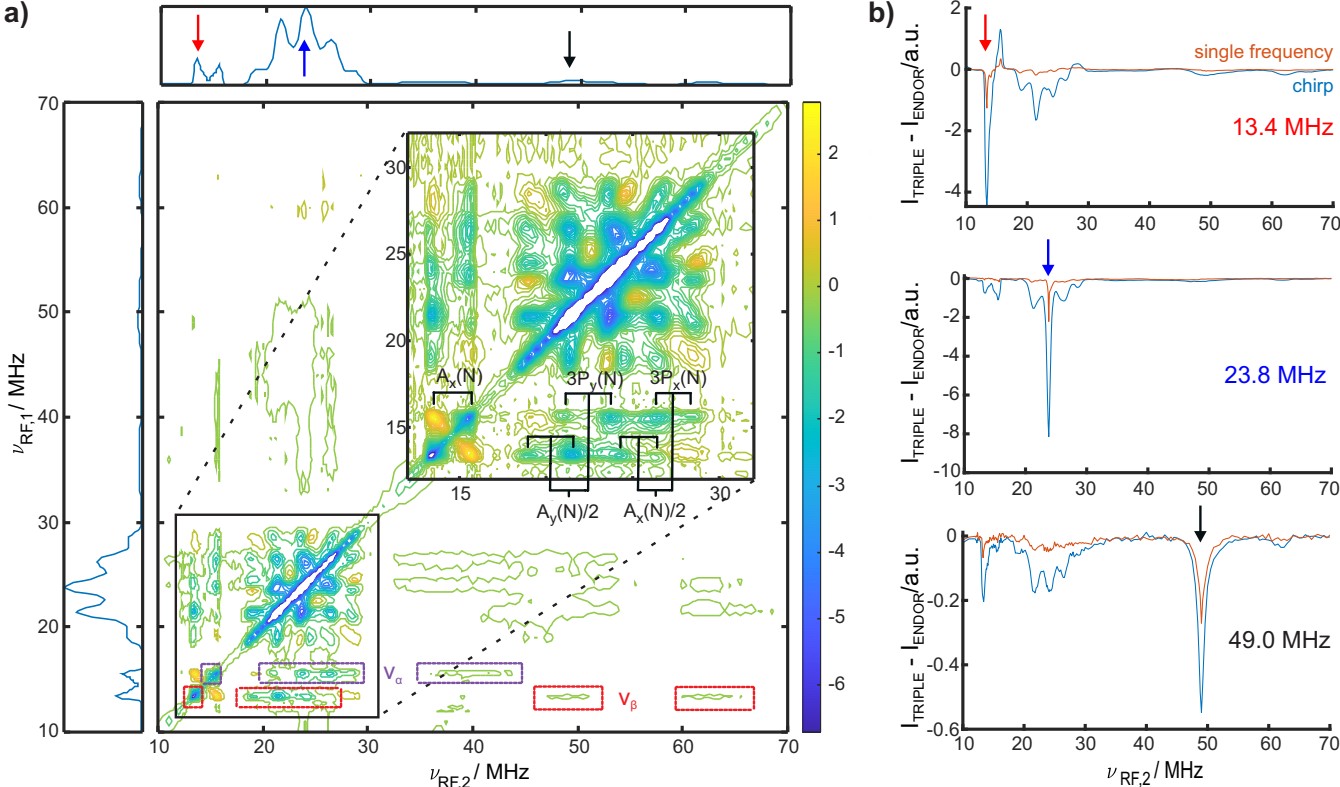

**Figure 4.** a) 2D TRIPLE difference spectrum of CuTPP with 40 µs chirped RF pulses with a bandwidth of 0.5 MHz. NMR peaks in the same electron spin manifold are marked in violet for $\nu_\alpha$ and red for $\nu_\beta$. The proton and nitrogen peaks are assigned and annotated according to spin Hamiltonian parameters given in Table S3. The ENDOR spectrum of CuTPP is shown as the projection along both axes. b) Comparison of TRIPLE difference traces at $\nu_{rf,1} = 13.4$ MHz ($^1$H), 23.8 MHz ($^{14}$N) and 49 MHz ($^{63,65}$Cu) with 40 µs chirped RF pulses versus 8 µs single frequency RF pulses (experimental optimal pulse length here for $^{14}$N). The acquisition time for the 2D TRIPLE was 2.7 days and for each 1D TRIPLE trace 30 minutes.

## 4   Conclusions

The substitution of the single frequency RF pulse by a chirped RF pulse can substantially increase the signal intensity especially for broad ENDOR lines, where we observed up to 9-fold increased intensity. The bandwidth of the chirp RF pulse offers the possibility for experimental optimization with respect to resolution and maximum signal intensity. In addition to Davies ENDOR experiments, chirp RF pulses can help to increase the sensitivity in different polarization-transfer ENDOR experiments that rely on the use of RF pulses, such as Mims ENDOR or TRIPLE experiments. The significant signal increase achievable by chirped RF excitation renders 2D experiments considerably more feasible on disordered samples, as demonstrated here by 2D TRIPLE with a 32.5-fold speed-up that brought acquisition time down to 2.7 days. The benefit of chirp ENDOR experiments

was shown at X-band frequencies, yet the RF chirp pulses are simpler to implement and less technically demanding compared to microwave pulses, and hence, these findings can easily be transferred to higher magnetic fields/frequencies with a suitable
arbitrary waveform generator as RF source.

*Acknowledgements.* JS, NW, and DK would like to acknowledge René Tschaggelar for helpful discussions on instrumentation.

*Code and data availability.* Data and data processing scripts are made available via Zenodo with the DOI 10.5281/zenodo.14039035.

*Author contributions.* JS and NW have carried out experiments and analyzed the data together with DK. JS, NW, NCN, and DK planned and organized different parts of the project. JS, NW, and DK designed the AWG-ENDOR setup. JS and DK wrote the manuscript with input
from all authors.

*Financial support.* Financial support from ETH research grant (ETH-35221) to DK is gratefully acknowledged. NCN and NW received financial support from the Aarhus University Research Foundation (Grant AUFF-E-2021- 9-22), the Swiss Natural Science Foundation (Postdoc.Mobility grant 206623), the Villum Foundation (Grant 50099), and the Novo Nordisk Foundation (Grant NNF22OC0076002).

*Competing interests.* The authors declare that they have no conflict of interest.

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
