# Peer review of "Increased sensitivity in Electron Nuclear Double Resonance spectroscopy with chirped radiofrequency pulses"

_Magnetic Resonance, 2024_

## Author Response (AR1)

**Response to the Reviewers' Comments on Manuscript mr-2024-14**

Here we address point by point each comment of the two Referees, and also for the community comment. The authors' replies are written in a blue normal font. Changes to the main text or the supporting information are marked below by a blue italic font and the corresponding line numbers are given for the marked-up manuscript. For the sake of clarity, all comments provided by the reviewers are numbered and indicated by a black normal font.

**RC1**: 'Comment on mr-2024-14', Anonymous Referee #1, 04 Oct 2024

The manuscript by Stropp et al reports ENDOR experiments using a chirped inversion pulse and are demonstrated on a model CuII-tetraphenylporphyrin complex in frozen solution. Hyperfine couplings in this molecular complex arise from protons, nitrogen of the ligands as well as the Cu(II) nucleus itself. Due to the intrinsic nature of the paramagnetic metal center, these couplings are anisotropic and spread out over tens of MHz. In the standard Davies and Mims ENDOR sequences, the ENDOR spectrum is probed stepwise (frequency domain) by a rectangular RF pulse. Replacing this pulse by a frequency-swept (or chirped) RF pulse, substantially increases the excitation bandwidth and results in a stronger ENDOR effect.

Chirp RF pulses in ENDOR have been introduced about three decades ago by Jeschke and Schweiger (1995) in the context of time-domain ENDOR experiments. Nevertheless, a demonstration in conjunction with the widespread frequency-domain experiment was somehow missed. This paper is now providing this information and also demonstrates the sensitivity gain but also the tradeoff with resolution for different types of hyperfine couplings. The experimental work is well-performed and complemented by more quantitative spin dynamics simulations. I can recommend publication after clarifying following points:

We thank the reviewer for the constructive feedback, which helps us to further improve our manuscript.

1. Page 3, phase cycle: I cannot find information on the phase cycle. Please explain better what kind of phases etc. are used.

   Thank you for noting this and we apologize for the missing description. The 4-step phase cycles used in the experiments are now included in section 2.2 of the revised manuscript:

   Line 92 - 94:
   *The phase cycle for Davies ENDOR was π ( 0, 0, 0, 0) – $p_{RF}$ (0, 0, 0, 0) – π/2 (0, 0, π, π) - π (0, π, 0, π) – Detection (1, 1, -1, -1) and for Mims π/2 ( 0, π, 0, π) – π/2 (0, 0, π, π) – $p_{RF}$ (0, 0, 0, 0) - π/2 (0, 0, 0, 0) – Detection (1, -1, -1, 1).*
   Line 101-102:
   *The phase cycling in TRIPLE experiments was the same as for Davies ENDOR.*

2. Page 7, Setting up the chirp pulse: I'm missing a discussion on how to set up or optimize the chirp inversion pulse. Lines 138-140 state that the performance depends on RF pulse power, pulse length and desired band width. However, the dependency on the inversion profile on these parameters is not discussed. Since this is the central part of the paper, a few more sentences would be desirable.

   Thank you for this important comment. As we fully agree with the reviewer that the experiments should be as accessible as possible, we include a new SI section "2.

Setup and Optimization of RF chirp pulses in ENDOR experiments" in the revised version of the manuscript:

*The first parameter to choose is the bandwidth of the chirp RF pulse and therefore the resolution desired in the experiment (e.g. 1 MHz chirp bandwidth corresponds to approximately 1 MHz potential resolution in the ENDOR spectrum as visible in Figs. 2 and 3a). As a second step the pulse length should be optimized such that the spectral power density is sufficient to achieve an adiabatic passage for spin packets with resonance frequencies within the excitation bandwidth. This is achieved when, the peak intensity does not increase anymore with increasing pulse length and depends on the available RF amplifier output power, the ENDOR resonator and the combined frequency response of the RF chain. In our case, for a 100 W RF amplifier output power and a Bruker X-band MD4 ENDOR resonator an RF pulse length of ca. 100 µs was sufficient for maximum sensitivity and full inversion of all coupled nuclei at 1 MHz chirp bandwidth (see Fig. 2d). We also recommend shaping the pulse edges with quarter sine waves to remove wiggles in the excitation profile. Even though plenty of information for optimal chirp RF pulses can be found in NMR literature; (Baum et al., 1985; Kupce and Freeman, 1996; Garwood and DelaBarre, 2001) the frequency response of the ENDOR RF circuit, which is often not known precisely and differences between spectrometer setups renders experimental testing of the optimal pulse parameters more straightforward and faster for many users than an optimization based on calculations.*

A reference to the SI section is made at the beginning of section 3.2 (line 163 - 164): *Details on how to set up and optimize chirp ENDOR experiments can be found in the SI section 2.*

3.  Page 7, lines 148 – 149: .. *2c and d show that the length of a 1 MHz chirp pulse does not have influence on the line intensity*… I'm confused by this statement as Fig. 2d) shows a clear dependence for 1H and 14N.

    We apologize for the ambiguous sentence. We wanted to convey that RF pulses longer than 100 µs do not change the ENDOR intensity anymore and are therefore not needed. We rephrased lines L171 – 175 accordingly:

    *Figures 2c and d show that the proton and nitrogen ENDOR sensitivity increases asymptotically to a maximum with longer chirp pulses. This can be explained by the slower frequency sweep rate through the fixed bandwidth of the chirp pulse for longer pulses, which enables a more adiabatic passage of the nuclear spins and a better inversion efficiency. (Baum et al. 1985, Doll et al. 2013) At a chirp bandwidth of 1 MHz a pulse length of about 100 µs is sufficient to achieve maximum intensity as visible in Fig. 2c and d.*

4.  Page 8, line 172: what is a 'mean' hyperfine coupling ? Please give the full tensor used in the simulation. What is the origin of the 1.2 MHz width (hyperfine anisotropy or convolution with a line width parameter) ?

    For the simulation a Gaussian distribution of purely isotropic hyperfine couplings was used. The maximum of this distribution $A_{mean}$ was set to 4 MHz. The width of this Gaussian distribution (FWHM) is 1.2 MHz corresponding to a standard deviation $\sigma$ of 0.5 MHz as introduced in methods section 2.3. We added the word *isotropic* where the hyperfine couplings are discussed and extended the description of the spin system in section 2.3 (line 107-110) by adding:

*The relative probability $p(A_i)$ of a specific hyperfine coupling $A_i$ is given by*

$$p(A_i) = \frac{1}{\sqrt{2\pi\sigma^2}} e^{-\frac{(A_i - A_{mean})^2}{2\sigma^2}} \qquad (1)$$

*The evolution of the spin density operator during the pulse sequence was simulated for each isotropic hyperfine coupling $A_i$ of the Gaussian distribution as an independent electron-nuclear 2-spin system.*

5. Simulation of the ENDOR spectra, Fig. 3B: The frequency domain spectrum is recorded by stepping the RF frequency. How is the convolution with the chirp pulse excitation profile performed ?

   We do agree with the reviewer that the text was not sufficiently detailed on this point, and updated the methods section 2.3 with a detailed description of the convolution (line 126 - 136):

   *The chirp ENDOR spectrum was further reproduced by convolution of an unbroadened, experimental single frequency (sf) spectrum and the RF chirp pulse excitation profile. The profile for each chirp bandwidth was calculated in EasySpin from the RF waveform shape (see SI section 2 for parameters of the waveform) for frequency offsets from the chirp center frequency $\nu - \nu_{RF}$. (Stoll and Schweiger, 2006; Pribitzer et al., 2016) A frequency-independent peak amplitude of 50 kHz was used, which corresponds to a sf $\pi$ pulse length of 10 µs. This excitation profile $E(\nu - \nu_{RF})$ was used in a discrete convolution with the 8 µs sf ENDOR spectrum $I_{sf}$ to obtain the broadened ENDOR spectrum $I_{sim,chirp}$:*

   $$I_{sim,chirp}(j) = (I_{sf} * E)(j) = \sum_{m=-M}^{+M} I_{sf}(j-m)E(m) \quad (5)$$

   *The discrete convolution requires that the spectrum and the excitation profile have the same frequency resolution, which was 0.1 MHz. Indices j, m and j − m indicate the position in the spectrum or excitation profile and ±M correspond to ±5 MHz offset in the excitation profile. After the convolution the simulated spectra for different chirp bandwidths were normalized and thus intensity-matched to the corresponding normalized chirp ENDOR spectrum.*

   We deleted the previous, shorter description, i.e. the following sentence in section 3.3 (line 220 - 221)

   *Each profile was calculated in EasySpin from the RF waveform shape and a frequency-independent peak amplitude of 50 kHz, which corresponds to a sf $\pi$ pulse length of 10 µs. (Stoll and Schweiger, 2006; Pribitzer et al., 2016)*

6. The reported TRIPLE ENDOR spectra are nice but in future it would be important to see a demonstration on a non-metal center. This type of experiment potentially suffers from T1n saturation as the same ENDOR transition is inverted/pumped at each step of the sequence. The nuclei close to a metal center might relax faster than in organic radicals, thus there might be a difference in performance.

   We agree that a demonstration on different systems (e.g. including organic radicals) will be useful to get a better understanding of the performance of TRIPLE on diverse paramagnetic systems. However, for the application class of metal sites (as, e.g., in catalysis), CuTPP is a relevant model system and hence a suitable test case. Thus, for applications in fields as catalysis, material science and bioinorganic chemistry, the chirp TRIPLE experiment on CuTPP in this paper showcases the utility of chirp

pulses in 2D experiments. We do, however, agree with the reviewer that on other systems such as slow-relaxing organic radicals, the performance of TRIPLE may well be somewhat worse, as may also be the case for TRIPLE without chirp pulses – however, this is beyond the scope of this paper. Note that to alleviate the influence of nuclear relaxation effects, stochastic RF excitation could be used, which in practice is often impractical for TRIPLE, yet might be helpful for observing slowly relaxing nuclei.

7. In conjunction with point (6), on page 2 line 7, the issues of nuclear saturation effects was reported in the paper by Rizzato et al, PCCP 2014 and not in Epel 2003. The latter discusses stochastic excitation for other reasons. This should be cited correctly.

We thank the reviewer for noting this incorrect citation, we corrected this in the revised manuscript (line 52).

**RC2**: 'Comment on mr-2024-14', Anonymous Referee #2, 19 Oct 2024

In this manuscript by Stropp et al. the application of chirped RF pulses in pulsed ENDOR and TRIPLE experiments on the example of Cu(II)-tetraphenylporphyrin is demonstrated. All experiments are in the frequency-domain, i.e. the spectra a recorded stepwise incrementing the rf frequency. Since the Cu-TPP complex in frozen solution features anisotropic hyperfine couplings with broad line in particular for $^{14}$N and $^{63,65}$Cu, through the introduction of a chirped rf pulse the authors can show a considerable intensity improvement in particular for the broader $^{14}$N and $^{63,65}$Cu lines. The authors also demonstrate the balance of intensity enhancement vs. spectral broadening by approaching the ENDOR line width with the bandwidth of the chirp pulse.

The experimental work is carefully performed and mostly well described. I can recommend publication after sorting out a few points:

We thank the reviewer for the positive assessment of our work and for the constructive feedback provided.

1. There is no EPR spectrum of Cu-TPP given, for completing the description at which B-field position the ENDOR/TRIPLE spectra are recorded there should be one displayed.
We added Fig. S1 with the experimental echo-detected field sweep to the SI along with tables S2 and S3 for the g-tensor and hyperfine & nuclear quadrupolar couplings known in literature:

*Table S2. g-tensor for CuTPP as published by Brown and Hoffman (1980).*

| Orientation | x | y | z |
|---|---|---|---|
| g | 2.045 | 2.045 | 2.190 |

*Table S3. Hyperfine couplings and nuclear quadrupolar couplings for CuTPP for the central Cu ion (in linear frequency units), the four chemically equivalent nitrogen atoms and pyrrole protons as published by Brown and Hoffman (1980).*

| Nucleus | $A_x$/MHz | $A_y$/MHz | $A_z$/MHz | $P_x$/MHz | $P_y$/MHz | $P_z$/MHz |
|---|---|---|---|---|---|---|
| $^{63}$Cu | -102.7 | -102.7 | -615 | | | |
| $^{14}$N | 54.213 | 42.778 | 44.065 | -0.619 | +0.926 | -0.307 |
| $^{1}$H | +2.5 | +0.70 | +0.80 | | | |

[Figure]

Figure S1. Echo-detected EPR spectrum of CuTPP at 9.78 GHz with an arrow marking the magnetic field position used for ENDOR experiments. For acquisition a Hahn echo sequence with 10/20 ns pulses and a τ of 420 ns was used with a 2-step phase cycle and a shot repetition time of 20 ms. The inset shows the structure of CuTPP.

2. The chirp ENDOR simulatins are not exhaustive explained enough, the zenodo link given in the previous reply to RC1 refers ot another manuscript.

We apologize for sharing the wrong zenodo link in the reply to Anonymous Referee 1 and in the manuscript. The correct zenodo doi is 10.5281/zenodo.14039035. We updated the manuscript in section 2.3 according to comment 4 & 5 of Anonymous Referee 1 (see lines 107 – 110, 126 - 136) with an extensive description of the ENDOR simulations. Additionally, we updated table S1 with the sampling rate and the vertical resolution of simulations and experiments. We also found a mistake in the simulation with the high 1 MHz RF chirp pulse power in Figure S7 (old S6) previously leading to unexpected asymmetries in the spectrum. The updated Figure S7 (before S6) now shows the new simulation with the correct sampling rate and without asymmetries, as expected in this case.

Table S1. Definition of parameters for chirp pulses and values used in simulations and experiments. *Amplitudes obtained from nutation experiments on $^{14}$N at 23.9 MHz.

| Parameter | Symbol | Value used in simulations | Value used in experiments |
|---|---|---|---|
| Center frequency | $\nu_{RF}$ | 4 - 24 MHz | 5 - 95 MHz |
| Bandwidth | $\Delta\nu_{chirp}$ | 0.062 - 8 MHz | 0.062 - 8 MHz |
| Amplitude* | $\nu_{2,max}$ | 100 & 1000 kHz | 140 kHz (500 W) & 62 kHz (100 W) |
| Pulse length | $t_{chirp}$ | 40 μs | 10 - 200 μs |
| Rise/Fall time | $t_{rise}$ | 0.2 μs | 0.2 μs |
| Sampling rate | $\nu_S$ | 2.4 GHz | 2.4 GHz |
| Vertical resolution | $r_v$ | 16 bit | 16 bit |

[Figure]

*Figure S7. Simulated chirp ENDOR spectra for different chirp bandwidths of an electron-proton 2-spin system with a Gaussian distribution of isotropic hyperfine couplings (σ = 0.5 MHz). The chirp pulse has length of 40 µs with 200 ns quarter sine wave weighted edges and an RF amplitude of $\nu_{2,max}$ = 1000 kHz.*

3. The loss in resolution in the chirp ENDOR spectrum when the bandwidth of the chirp rf pulse approaches the ENDOR linewidth can be understood similar to "overmodulating" a line in cw-EPR, spectral contributions add then to the line intensity which don't belong to the center rf frequency of the pulse. Unfortunately the authors stop their chirp ENDOR simulations at more interesting cases: [14]N ENDOR lines with NQI, where the chirp rf pulse can excite adjacent transitions in the same $m_s$ manifold. The authors should explain, why the simulations become then "infeasible".

What we meant with the simulations "become infeasible" is that the computational cost to perform time domain simulations for spin systems with a higher number of spins increases dramatically, thus hampering spectral analysis by least-squares fitting of spin Hamiltonian parameters. This is the reason why we think the convolution approach is of interest for the typical ENDOR spectroscopist: One can still use frequency domain simulations (as implemented in EasySpin) to analyze the experimental chirp ENDOR spectra as described.
We kept the spin dynamics simulations simplistic on purpose, because they should be easy to understand and explain the basic working principle of chirp pulses in ENDOR. We agree that chirp pulses open up exciting new possibilities and should be studied in more detail (e.g. for lines of nuclei with I>1/2 and quadrupolar coupling). In our opinion, this would benefit from a more thorough in-depth investigation rather than being a sidenote in this experimentally focused method paper on ENDOR.

We extended the explanation to clarify what we mean by "infeasible" in more detail in the revised manuscript in line 216 - 218:

*Spin dynamics simulations provide valuable insights into the chirp ENDOR experiment, but they become infeasible for spectral analysis of larger spin systems as in CuTPP due to the dramatic increase in computational cost for time-domain simulations with increasing numbers of spins, (Kuprov et al., 2007) and a simpler simulation approach becomes necessary.*

Regarding the question on 14N & NQI with respect to why a convolution becomes infeasible at large RF chirp bandwidths for such cases, we updated the explanation in lines 224 – 229 and included two references:

Second, the chirp pulse might affect multiple ENDOR transitions in the same electron spin manifold, which is especially relevant here for I > ½, as for $^{14}$N. Transitions that have an energy level in common will interfere with each other during the passage of the nuclear spin transitions by the chirp RF pulse. (Doll and Jeschke, 2017; Jeschke et al., 2015) In such cases, the convolution approach is not expected to result in an accurate line shape.

4. l178-184: The authors mix here chirp rf pulses (with too large bandwidth) with special TRIPLE experiments, which are not comparable: the chirp ENDOR are broadened beyond the overall spectral width, whereas in the special TRIPLE case (with small bandwidth) the 2 NMR transitions are hit simultaneously inreasing sensitivity and enabling quantitative experiments.

We apologize for not writing this clearly enough. We did not to compare chirp ENDOR with special TRIPLE experiments, but just point out to the reader that in contrast to the chirp ENDOR experiments, in special TRIPLE experiments the consecutive excitation of 2 NMR transitions is exploited in a favorable, quantitative fashion. We now clarify this by rewriting the sentences in section 3.3 (line 211 – 215):

*In contrast to the 4 and 8 MHz chirp ENDOR spectra here, the excitation of both coupled NMR transitions has been exploited before with two separate single frequency RF pulses in special TRIPLE experiments in a favorable, quantitative fashion. (Dinse,1974; Epel,2003) In chirp ENDOR the double excitation with a single chirp RF pulse is unwanted since it complicates the spectrum and should be avoided.*

5. l155: The discussion of overtones from the rf amplifier leading to artificial lines in the ENDOR spectrum should be taken with care. Usually the occurance of higher harmonics of an amplifier is specified at 0 dBm rf input. Working within this limit or beyond is in the hands of the operator.

We agree that it is in the hands of the operator to adjust the RF power carefully. Note however, that the intensity of visible overtones also depends on the extend of spectral averaging. We updated section 3.2. with a warning about these overtones (line 183 – 186):

*These artificial lines from higher harmonics of the RF amplifier output occur at high RF powers, when the shortest RF pulses are used for maximum ENDOR sensitivity. They are best avoided, since they may corrupt the ENDOR spectrum. To this end, chirp ENDOR experiments deliver the possibility to reduce RF power without sacrificing sensitivity.*

6. When it comes to the presentation of the TRIPLE results, it would be more insightful, if the authors not only explain how the spectra were acquired, but also how they can be interpreted. In this regards I can only emphasize the need of tabularised hyperfine couplings and their signs for Cu-TPP. This would make these experiments more attractive for readers,

who are aiming for this information on other systems. In this context fig.4a stands rather unexplained, and the reader doesn't really know, how to interpret the 2D TRIPLE experiment (and if its worth to perform it for 2 days rather than a few 1D TRIPLE traces at specific hfc's).

We now include a table with hyperfine & nuclear quadrupolar couplings for Cu-TPP in the SI of the manuscript (Tables S2 & S3, see also answer to reviewer 2 comment 1). Further, we annotated the 2D TRIPLE spectrum in Fig. 4a with hyperfine and nuclear quadrupolar couplings as far as they are known and resolved and modified the figure caption accordingly.
We agree with the reviewer that for CuTPP, it is enough to record 1D TRIPLE traces at a few hfcs to read out the same information as from the 2D TRIPLE. In this case, the average saving in measurement time of 95 % (specific for the pulse parameters & sample in this experiment) holds true for both 1D TRIPLE traces and 2D TRIPLE, since both rely on 2 RF pulses. However, the 2D TRIPLE serves as a demonstration for more complicated systems with multiple paramagnetic centers, where 2D TRIPLE is required for a clear interpretation and we show (although on the comparably simple CuTPP system) that chirp RF pulses are valuable also for such cases.

[Figure]

*Fig. 4 a) 2D TRIPLE difference spectrum of CuTPP with 40 µs chirped RF pulses with a bandwidth of 0.5 MHz. NMR peaks in the same electron spin manifold are marked in violet for $v_\alpha$ and red for $v_\beta$. The proton and nitrogen peaks are assigned and annotated according to spin Hamiltonian parameters given in Table S3. The ENDOR spectrum of CuTPP is shown as the projection along both axes. b) Comparison of TRIPLE difference traces at $v_{rf,1}$ = 13.4 MHz ($^1$H), 23.8 MHz ($^{14}$N) and 49 MHz ($^{63,65}$Cu) with 40 µs chirped RF pulses versus 8 µs single frequency RF pulses (experimental optimal pulse length here for $^{14}$N). The acquisition time for the 2D TRIPLE was 2.7 days and for each 1D TRIPLE trace 30 minutes.*

We added an explanation of the assignment in section 3.4 (lines 250 – 255):

*In the 2D spectrum the NMR transitions in the same electron spin manifold can be*

*clearly identified for all three coupled types of nuclei. This separation reduces the number of peaks along one dimension by a factor of 2 and the eight overlapping nitrogen NMR transitions are resolved and can be assigned (see zoom inset in Fig. 4a) For this purpose a single chirp RF pulse should not excite two RF transitions from different electron spin manifolds, which is why chirp bandwidth of 0.5 MHz was chosen as a compromise between gain in signal intensity and necessary resolution.*

7. In fig.4b the 1D difference TRIPLE traces and the improvement of intensity are shown, the measurement time would be interesting here, too, to compare with 2D TRIPLE and ENDOR.

   A direct comparison of the signal increase for 2D TRIPLE with respect to ENDOR for a certain measurement time is difficult since a slightly different number of points and non-uniform sampling were used. In our opinion, the difference between chirp and standard single frequency TRIPLE/ENDOR is more important, since the cases of usage are slightly different, and we focused on this comparison. We now include the measurement time (30 min / 1D TRIPLE trace) in the caption of Fig. 4:

   *The acquisition time for the 2D TRIPLE was 2.7 days and for each 1D TRIPLE trace 30 minutes.*

8. For the TRIPLE experiments the bandwidth-dependent intensity/broadening effects are not unfortunately not investigated, which are to be expected in favour of intensity enhancement:
   Since anisotropic lines are excited with nu(rf1), the higher the fraction of the excited anisotropic hyperfine line, the higher the intensity effect should be. Have the authors this possible effect taken into consideration, or even investigated?

   Thank you for the comment. We expect that the bandwidth-dependent effects in TRIPLE will be similar to ENDOR results as long as the chirp pulse does not excite multiple transitions from different electron spin manifolds, and we therefore did not investigate this further. We chose to use a chirp pulse bandwidth of 0.5 MHz, since this bandwidth is small enough to excite only single nitrogen lines and therefore gives the best sensitivity gain with respect to this frequency region. We agree that for 2D TRIPLE with maximum sensitivity the chirp pulse bandwidth should be adjusted to the width of the anisotropic line, as we already note in line 229, the chirp bandwidth could even be dynamically adapted specifically to each ENDOR line, resulting in a non-uniform bandwidth excitation scheme. We added the following sentences at the end of section 3.4 (line 273 – 276):

   *As an optimal reference for this case, the ENDOR spectrum will then also be measured with the same non-uniform bandwidth excitation scheme as the TRIPLE to obtain a well-defined TRIPLE difference spectrum. While technically feasible, the quantitative information of peak intensities among different coupled nuclei in the ENDOR spectra might become compromised, which remains to be tested in further studies.*

**CC1**: , Fabian Hecker, 02 Oct 2024

The paper discusses the use of chirped RF pulses in ENDOR spectroscopy of a frozen solution transition metal complex model system at X-band frequencies. It emphasizes the significant sensitivity enhancements provided by chirped pulses, which are particularly notable for broad ENDOR lines associated with nuclei having spin I > 1/2 and transition metal nuclei. The authors carefully examine the trade-off between increased sensitivity and line broadening when the chirp bandwidth approaches or exceeds the linewidth. Additionally, they demonstrate how this sensitivity improvement enables multidimensional ENDOR experiments, such as TRIPLE, to be conducted within practical time frames—overcoming a major limitation that has hindered the adoption of these techniques since their development. There are a few points that benefit from clarification:

We thank Fabian Hecker for taking the time to comment on our paper and provide positive and constructive feedback. To adequately take these suggestions into account, we include them in the present answers.

Line 32: The authors state that Davies does not suffer from blind spots. While it may not exhibit periodic blind spots, the technique does suffer from a central blind spot at the nuclear Larmor frequency, which is determined by the excitation pulse width. Although this may not be significant in the case discussed, it often has a considerable impact on the analysis of small hyperfine couplings.

We updated the following sentence in the introduction (line 33 – 35):

*Of the two most widely applied pulse ENDOR experiments, Davies ENDOR (Davies 1974) is most suited for this purpose, since it only features a central blind spot at the nuclear Larmor frequency. This diminishes the intensity of very small hyperfine couplings, but otherwise does not significantly distort the line shapes of peaks from nuclei with stronger hyperfine couplings.*

Line 120: CuTPP is discussed as a well-known model system. Consequently, the hyperfine couplings should be provided here to facilitate evaluation of the spectra.

We included tables S2 & S3 with g-tensor, hyperfine and quadrupole couplings of the relevant coupled nuclei observed in the ENDOR experiments to the supporting information (see reviewer 2 comment 1).

Figure 2:

- The use of 500 W to achieve maximum RF power is understandable; however, since all other comparisons are made to 100 W spectra, it might be advisable to omit the 500 W data from the figure to avoid confusion.

  We would prefer to keep the 500 W single frequency ENDOR spectrum in the figure to clearly show that higher sensitivity can be achieved with much lower power by chirp RF pulses.

- It is unclear whether integrated signals or peak intensities are being discussed in panels b) and d).

  Peak intensities are compared in these panels. We added the following sentence to the methods section 2.3 (line 95 - 96):

*For comparison of different ENDOR intensities, peak intensity values at different spectral positions were extracted after this offset correction.*

- The color code in panel b) is somewhat confusing, as red represents ¹H and blue represents ¹⁴N, but pale blue is used for ¹H and pale red for ¹⁴N.

  We apologize that the position/color of the pale arrows was swapped in Figure 2b and we have updated this now.

Line 186 and Figure 3: The convolution of the experimental single-frequency ENDOR spectrum is a clever method for analyzing the effect of the chirp pulse. This suggests that the same analysis could be achieved with a standard frequency-domain simulation of the spectrum, potentially improving the interpretation of the ENDOR spectra without requiring a dedicated spin dynamics simulation.

Thank you for this comment. As noted, we show in Fig. 3 that a convolution-based approach is valid to simulate the chirp ENDOR spectrum. This is most apparent by comparing the experimental chirp ENDOR spectra to a convoluted experimental single frequency spectrum rather than to a convoluted frequency-domain ENDOR simulation. This provides the most accurate comparison by overlaying experimental data with RF-excitation width broadening together with single-frequency data broadened in post processing. Accordingly, a frequency domain simulation, which describes the single frequency spectrum well, will also describe the chirp ENDOR spectrum well after convolution (note, not necessarily the other way around).

To strengthen this point in the revised manuscript, we emphasize these points in section 3.3 (line 232 – 235):

*This shows that chirp ENDOR spectra can be analyzed using frequency-domain simulations to obtain the unbroadened spectrum and subsequent convolution with the chirp pulse excitation bandwidth to compute the experimental broadened spectrum. Fitting spin Hamiltonian parameters using chirp ENDOR spectra is thereby feasible in a manner analogous to using single frequency ENDOR data.*

Technical: Line 130:  \mu s instead of \muand

Thank you for noticing the typo.